# SINGLE-PHOTON IMAGE CLASSIFICATION

## ABSTRACT

Quantum Computing based Machine Learning mainly focuses on quantum computing hardware that is experimentally challenging to realize due to requiring quantum gates that operate at very low temperature. We demonstrate the existence of a "quantum computing toy model" that illustrates key aspects of quantum information processing while being experimentally accessible with room temperature optics. Pondering the question of the theoretical classification accuracy performance limit for MNIST (respectively "Fashion-MNIST") classifiers, subject to the constraint that a decision has to be made after detection of the very first photon that passed through an image-filter, we show that a machine learning system that is permitted to use quantum interference on the photon's state can substantially outperform any machine learning system that can not. Specifically, we prove that a "classical" MNIST (respectively "Fashion-MNIST") classifier cannot achieve an accuracy of better than $22.96\%$ (respectively $21.38\%$ for "Fashion-MNIST") if it must make a decision after seeing a single photon falling on one of the $28 \times 28$ image pixels of a detector array. We further demonstrate that a classifier that is permitted to employ quantum interference by optically transforming the photon state prior to detection can achieve a classification accuracy of at least $41.27\%$ for MNIST (respectively $36.14\%$ for "Fashion-MNIST"). We show in detail how to train the corresponding quantum state transformation with TensorFlow and also explain how this example can serve as a teaching tool for the measurement process in quantum mechanics.

## 1 INTRODUCTION

Both quantum mechanics and machine learning play a major role in modern technology, and the emerging field of AI applications of quantum computing may well enable major breakthroughs across many scientific disciplines. Yet, as the majority of current machine learning practitioners do not have a thorough understanding of quantum mechanics, while the majority of quantum physicists only have an equally limited understanding of machine learning, it is interesting to look for "Rosetta Stone" problems where simple and widely understood machine learning ideas meet simple and widely understood quantum mechanics ideas. It is the intent of this article to present a setting in which textbook quantum mechanics sheds a new light on a textbook machine learning problem, and vice versa, conceptually somewhat along the lines of Google's TensorFlow Playground (Smilkov et al. (2017),) which was introduced as a teaching device to illustrate key concepts from Deep Learning to a wider audience.

Specifically, we want to consider the question what the maximal achievable accuracy on common one-out-of-many image classification tasks is if one must make a decision after the detection of the very first quantum of light (i.e. photon) that passed a filter showing an example image from the test set. In this setting, we do not have a one-to-one correspondence between example images from the training (respectively test) set and classification problems. Instead, every example image defines a probability distribution for the $(x, y)$ detector pixel location on which the first photon passing an image filter lands, the per-pixel probability being the pixel's brightness relative to the accumulated (across all pixels) image brightness. So, from every ($28 \times 28$ pixels) example image, we can sample arbitrarily many photon-detection-event classifier examples, where the features are a pair of integer pixel coordinates, and the label is the digit class.

On the MNIST handwritten digit dataset (LeCun and Cortes (2010)), any machine learning system that only gets to see a single such "photon detected at coordinates $(x, y)$" event as its input features, of the pixel that flashed up are the only input features, is limited in accuracy

by the maximum likelihood estimate, since we have: $P(\text{Image class C}|\text{Photon detected at }(x, y)) = \sum_{\text{E}} P(\text{Image class C}|\text{Example E})P(\text{Example E}|\text{Photon detected at }(x, y))$.

On photon detection events generated each by first randomly picking an example image, and then randomly picking a brightness-weighted pixel from that, we cannot do any better than predicting the most likely digit class given these input features – the two pixel coordinates. As performance is measured on the test set, no classifier could possibly ever outperform one that is built to achieve maximal performance on the test set. This is obtained by determining, for each pixel, what the most likely class is, where examples from the test set are weighted by the fraction of total example-image brightness that comes from the pixel in question. Figure 2(b) shows the most likely image-class per pixel. (For MNIST, some pixels are dark in every test set example.) No classifier can outperform one that simply looks up the pixel-coordinates at which a photon was detected in Figure 2(b) and returns the corresponding class, and this optimal classifier's accuracy is $22.96\%$ for the the MNIST dataset – substantially higher than random guessing ($10\%$). Appendix A.2 provides a detailed (but mostly straightforward) optimality proof of this accuracy threshold. We cannot, for example, outperform it by redistributing light intensity between pixels, since any such redistribution could only destroy some of the available useful information, not magically create extra useful information.

An entirely different situation arises when we allow quantum mechanics to enter the stage: For a single photon passing through a coherently illuminated image filter, with all pixels at the same optical phase on the incoming wave, we can imagine putting some precision optical device between the image filter and the detector array that redistributes not the *probabilities* (which correspond to light intensity when aggregating over many photons), but the *amplitudes* that make up the spatial part of the photon wave-function. Illuminating such a set-up with many photons would show a hologram-like interference pattern on the detector array. This transformation of the (single-)photon wave function by linear optical elements then has tuneable parameters which we can adjust to improve classifier accuracy. Quantum mechanics tells us that every (lossless) linear optical device can be represented by a linear unitary transform on the photon state: The action of any complex optical device consisting of (potentially very many) components which transforms a $N$-component photon state (in our case, $N = 28^2$ amplitudes in the spatial part of the photon wavefunction) can be described by an element of the $N^2$-dimensional unitary matrix Lie group $U(N)$. Vice versa, Reck et al. (1994) describes a constructive algorithm by which any $U(N)$ transformation matrix can be translated back to a network of optical beam splitters and phase shifters.

## 1.1 RELATED WORK

Conceptually, exploiting interference to enhance the probability of a quantum experiment producing the sought outcome is the essential idea underlying all quantum computing. The main difference between this problem and modern quantum computing is that the latter tries to perform calculations by manipulating quantum states of multiple "entangled" constituents, typically coupled two-state quantum systems called "qubits," via "quantum gates" that are controlled by parts of the total quantum system's quantum state. Building a many-qubit quantum computer hence requires delicate control over the interactions between constituent qubits. This usually requires eliminating thermal noise by going to millikelvin temperatures. For the problem studied here, the quantum state can be transformed with conventional optics at room temperature: the energy of a green photon is 2.5 eV, way above the typical room temperature thermal radiation energy of $kT \simeq 25$ meV. The price to pay is that it is challenging to build a device that allows multiple photons to interact in the way needed to build a many-qubit quantum computer. Nevertheless, Knill, Laflamme, and Milburn (Knill et al. (2001)) devised a protocol to make this feasible in principle, avoiding the need for coherency-preserving nonlinear optics (which may well be impossible to realize experimentally) by clever exploitation of ancillary photon qubits, boson statistics, and the measurement process. In all such applications, the basic idea is to employ coherent multiphoton quantum states to do computations with multiple qubits. In the problem studied here, there is only a single photon, the only relevant information that gets processed is encoded in the spatial part of its wave function (i.e. polarization is irrelevant), so the current work resembles the "optical simulation of quantum logic" proposed by Cerf et al. (1998) where a N-qubit system is represented by $2^N$ spatial modes of a single photon. Related work studied similar "optical simulations of quantum computing" for implementing various algorithms, in particular (small) integer factorization (Clauser and Dowling (1996); Summhammer (1997)), but to the best of the present authors' knowledge did not consider machine learning problems.

This work can be described as belonging to the category of machine learning methods on quantum non-scalable architectures. Alternatively, one can regard it as a quantum analogue of recent work that demonstrated digital circuit free MNIST digit classification via classical nonlinear optics, for instance via saturable absorbers (Khoram et al. (2019).) Apart from providing an accessible and commonly understandable toy problem for both quantum and ML research communities, this simple-quantum/simple-ML corner also may be of interest for teaching the physics of the measurement process ("the collapse of the wave function") in a more accessible setting. Whereas explanations of the measurement process are forced to remain vague where they try to model "the quantum states of the observer" (typically unfathomably many states that one would never hope to be able to model in terms of actual numbers), using machine learning as a sort-of cartoon substitute for high level mental processes actually allows us to come up with fully concrete toy models of the measurement process on low-dimensional (such as: $D < 1000$) Hilbert spaces that nevertheless capture many of the essential aspects – to the extent that "ML classifies the measurement as showing the image of a shoe" can be regarded as a crude approximation to "observer sees a shoe".

Looking closer at the relation between the present article and Khoram et al. (2019), both articles study the general feasibility of realizing Machine Learning classifiers in the form of an analog optical computer at the theoretical level, using numerical optimization to produce a blueprint of a device that can perform inference for a specific problem. In both articles, the primary problem under study is MNIST handwritten digit classification, the input is encoded as spatial dependency of a (monochromatic) laser beam's light intensity, and classification happens by using interference to funnel optical energy onto a detector array. In both cases, stochastic gradient descent is used to shape how this funneling of optical energy happens. Indeed, even the loss function used for training (cross entropy) is essentially equivalent. The key differences are that Khoram et al. (2019) only considers the many-photon limit of classical wave optics, which allows one the luxury of using non-linear optical components, specifically saturable absorbers, to implement non-linearities. This has no analog for the single photon case. Also, having many photons available allows identifying the target class that receives most laser light and calling this the prediction of the model. This is clearly not possible when a decision has to be made after seeing only a single photon. If one sent many photons through an interference device as described in this article and picked the target class with the highest photon count, one would observe classification accuracies of about 90% rather than the claimed about-40% for a single photon. This is considerably higher than the accuracies of about 80% presented inKhoram et al. (2019) as the focus of that article is on manufacturability, running gradient backpropagation directly on a Finite Difference Frequency Domain PDE simulation of Maxwell's equations and taking materials engineering constraints into account, whereas our work focuses on upper and lower bounds for achievable accuracy, exploiting the one-to-one equivalence between linear optical devices and unitary transforms. Our work directly trains the parameters of the unitary transform, which only afterwards get mapped to a blueprint for an experimental device realization. Speculatively, if a device were built experimentally that was designed by the methods inKhoram et al. (2019), subject to the extra constraint that no non-linear elements can be used, and then deployed in a low-light-intensity single-photon setting, using a suitable detector such as a SPAD array, it may manage to realize better-than-classically-achievable classifier performance, for reasons explained in the current work.

## 1.2 THE MEASUREMENT PROCESS

How well one can one solve a mental processing task, such as identifying a handwritten digit, if one is permitted to only measure a single quantum? This question leads to a Hilbert space basis factorization that parallels the factorization needed to study the quantum mechanical measurement process. Let us consider a gedankenexperiment where our quantum system (see Feynman et al. (2010); Landau and Lifshitz (1981) for an introduction to quantum mechanics) is a single atom that has two experiment-relevant quantum states, 'spin-up' and 'spin-down',

$$|\psi_{\text{Atom}}\rangle = c_0|\psi_{\text{Atom}=\uparrow}\rangle + c_1|\psi_{\text{Atom}=\downarrow}\rangle. \tag{1}$$

This atom undergoes a measurement by interacting, over a limited time period, with an apparatus. The measurement process may involve for instance an atom emitting a photon that is detected by a camera, and it may include a human observing the result. We describe a quantum state in the potentially enormous Hilbert space of apparatus states with the vector $|\psi_{\text{Apparatus}}\rangle$. If, in this gedankenexperiment, we actually assume that we have maximal information about the (quantum)

state of the measurement apparatus (which, however, in practical terms would be unfathomably complicated) at the beginning of the experiment, then the full quantum state of the initial system is the tensor product

$$|\psi_{\text{System, initial}}\rangle = |\psi_{\text{Atom, initial}}\rangle \otimes |\psi_{\text{Apparatus, initial}}\rangle. \qquad (2)$$

This factorization implies that atom and apparatus states are independent before the interaction.

Without interaction between the apparatus and the atom, the time-evolution of the total system factorizes. A measurement requires an interaction between the apparatus and the atom, the solution of the Schrödinger equation is equivalent to the application of a unitary operator $U$ to the state $|\psi_{\text{System, initial}}\rangle$. This has the effect of combining the state components of the atom and the apparatus and, as a consequence, the joint time evolution no longer can be factorized. The overall state is $|\psi_{\text{System, final}}\rangle = U|\psi_{\text{System, initial}}\rangle$ and can be always decomposed in the sum:

$$|\psi_{\text{System, final}}\rangle = \alpha|\psi_{\text{Atom}=\uparrow}\rangle \otimes |\psi_{\text{Apparatus, final}=\uparrow}\rangle + \beta|\psi_{\text{Atom}=\downarrow}\rangle \otimes |\psi_{\text{Apparatus, final}=\downarrow}\rangle, \qquad (3)$$

where the apparatus states $|\psi_{\text{Apparatus, final}=\uparrow}\rangle$ and $|\psi_{\text{Apparatus, final}=\downarrow}\rangle$ represent the state of the apparatus after the measurement for the two basis states of the atom. Therefore, the apparatus is in a different state for the two cases, which leads to the apparent "collapse" of the wave function. The apparatus in the state $|\psi_{\text{Apparatus, final}=\uparrow}\rangle$ perceives the "collapse" because the atom seems to have taken the state $|\psi_{\text{Atom}=\uparrow}\rangle$. The state of the apparatus includes also the representation of the thought process of a possible human observer, for instance asking herself at what instant the atom took a well determined state. This thought process disregards the superposed state $|\psi_{\text{Apparatus, final}=\downarrow}\rangle$ which represents the alternative reality, where the apparatus observed a different outcome.

Considering that a mental process could be seen as a measurement on the environment, one would naturally be inclined to think that high level mental concepts never would naturally lend themselves to a description in terms of some Hilbert space basis that has tensor product structure $|\psi_{\text{general concept}}\rangle \otimes |\psi_{\text{details}}\rangle$. Machine learning is now making the question to what extent this may nevertheless work quantitatively testable for some simple cases, if we consider it as providing reasonably good (for this purpose) models for mental concepts.

Let us consider the spatial part of a single photon's quantum state as it traveled through a mask that has the shape of a complicated object. For instance, let's assume that the mask is obtained from a random sample of the "Fashion-MNIST" dataset, Xiao et al. (2017), where each sample represents an object such as a shirt, a trouser, etc. One would generally expect that any sort of transformation that connects a highly regular and mathematically simple description of such a quantum system, such as in terms of per-picture-cell ("pixel") amplitudes, with a description in human-interpretable terms, such as "the overall intensity pattern resembles a shirt," would unavoidably involve very complicated entanglement, and one should not even remotely hope to be able to even only approximately express such photon states in terms of some factorization

$$|\psi_{\text{photon}}\rangle \approx \sum_{\text{shape class } C \in \{shirt, trouser, ...\}} \sum_{\text{style } S} c_{CS}|\psi_{\text{shape class } C}\rangle \otimes |\psi_{\text{style } S}\rangle, \qquad (4)$$

since one would not expect the existence of a basis of orthonormal quantum states that can be (approximately) labeled $|\psi_{\text{shirt}}\rangle$, $|\psi_{\text{shoe}}\rangle$, etc. Using machine learning, we can quantitatively demonstrate that, at least for some simple examples, *precisely such a factorization does indeed work remarkably well*, at least if we content ourselves with the concept of a "shirt shape" being that of a one-out-of-many machine learning classifier, so not quite that of a human. In any case, it is reassuring to see that even near-future few-qbits quantum computers might be able to model high level concepts rather well.

## 2 SINGLE-QUANTUM OBJECT CLASSIFICATION

Our gedankenexperiment starts with a single photon passing from faraway through a programmable LCD screen, which we here consider to consist of $N \times N$ pixels and show an image, where for both the MNIST handwritten digit dataset of LeCun and Cortes (2010) and the "Fashion-MNIST" dataset of Xiao et al. (2017), we have $N = 28$.

The size of the screen shall be sufficiently small for the photon's quantum state to be at the same phase as it reaches each individual pixel. This does not mean that the screen has to be small in comparison to the wavelength. Rather, the light source must provide highly collimated illumination.

The relevant spatial part of the photon's quantum state is described by an element of a $N \times N$-dimensional complex vector space. We can choose a basis for this Hilbert space such that the quantum state of a photon that managed to pass through the screen (rather than getting absorbed) has the form

$$|\psi_{\text{Photon}}\rangle = \sum_{\text{row } j, \text{ column } k} c_{jk}|\psi_{jk}\rangle \tag{5}$$

where the $|\psi_{jk}\rangle$ basis functions correspond to a photon that went through pixel $(j,k)$, and the coefficients $c_{jk}$ are real, non-negative, proportional to the square roots of the image's pixel-brightnesses, and are normalized according to $\sum_{j,k}|c_{j,k}|^2 = 1$.

As we want to perform a rotation on this Hilbert space that maximizes alignment with a tensor product Hilbert space where one factor describes an image class, we pad this $N^2$-dimensional Hilbert space into a larger Hilbert space with dimensionality $M$ divisible by the number of object classes $C$, i.e. $M = C \cdot S$. This amounts to adding always-dark pixels (that may not form a complete row) to the image. The problem then amounts to engineering, for a problem $P$ such as handwritten digit recognition, a single problem-specific unitary transform $U_P$ of the photon state, $|\psi_{\text{Photon}}\rangle \to U_P|\psi_{\text{Photon}}\rangle$, such that we can meaningfully claim:

$$U_P|\psi_{\text{Photon}}\rangle = |\psi_{\text{Photon}^*}\rangle \approx \sum_{\text{example class } c} \sum_{\text{style } s} c_{cs}|\psi_{\text{class is } c}\rangle \otimes |\psi_{\text{style variant is } s}\rangle \tag{6}$$

Specifically, for each individual example image $E$, we would like to have

$$U_P|\psi_{\text{Photon},E}\rangle \approx |\psi_{C(E)}\rangle \otimes \sum_{\text{style } s} c_s|\psi_{\text{style variant is } s}(E)\rangle, \tag{7}$$

where $C(E)$ is the ground truth label of the example in a supervised learning setting.

Using the method described in Reck et al. (1994), this trained matrix then can be translated to an optical network blueprint. The transformed quantum state at the output side of the network of beam splitters and phase shifters then gets measured by a detector array that can discriminate $M = C \cdot S$ quantum states which are labeled $|\psi_{\text{digit is a } 0}\rangle \otimes |\psi_{\text{style variant } 1}\rangle$, $|\psi_{\text{digit is a } 0}\rangle \otimes |\psi_{\text{style variant } 2}\rangle$, ..., $|\psi_{\text{digit is a } 3}\rangle \otimes |\psi_{\text{style variant } 57}\rangle$, ..., $|\psi_{\text{digit is a } 9}\rangle \otimes |\psi_{\text{style variant } S_{\max}}\rangle$. If we detect the photon in any of the $|\psi_{\text{digit is a } 7}\rangle \otimes \ldots$ cells, the classifier output is a "7", and likewise for the other digits.

From a machine learning perspective, the trainable parameters hence are the complex entries of the matrix $U_P$, which according to quantum mechanics have to satisfy an unitarity constraint, $U_P U_P^\dagger = I$, and this search space automatically covers all experimentally realizable linear optical devices. For MNIST, where examples have $28 \times 28$ pixels, the most obvious choice is padding to a $M = 790$-dimensional input vector. While one could implement the unitarity constraint in terms of a (regularizer) loss-function contribution that measures the degree of violation of unitarity, it here makes more sense to instead use a parametrization of $U_P$ that automatically guarantees unitarity, using Lie group theory. If $W_P$ is a $790 \times 790$ matrix of trainable (real) weights, then the hermitean matrix $H_P = -i(W_P - W_P^T) + (W_P + W_P^T)$ parametrizes the Lie algebra $\mathfrak{u}(790)$, and $U_P = \exp(iH_P)$ covers all of the (compact) unitary group $U(790)$. This approach slightly over-parametrizes the problem, since, in the tensor-product basis that we are transforming to, we can freely re-define the basis on each of the ten $790/10 = 79$-dimensional style subspaces. This means that 10% of the parameters are redundant.

Overall, with all the trainable weights being provided by the matrix $W_P$, and the brightness of the pixel at coordinates $(y, x)$ for example $E$ being $b_{E;yx}$, we have this formula for the probability of a photon travelling through an optical device that was designed by training weights and landing on a detector cell that predicts class $c$:

$$p(c|E) = \sum_s \left| \sum_{j,k,y,x} \text{expm}\left(W_P - W_P^T + i(W_P + W_P^T)\right)_{kj} \sqrt{\frac{b_{E;yx}}{\sum_{\tilde{y},\tilde{x}} b_{E;\tilde{y}\tilde{x}}}} \delta_{N \cdot y + x, j} \delta_{j, c \cdot S + s} \right|^2 . \tag{8}$$

Here, $y$ and $x$ are image row- and column-indices (for MNIST, running from 0 to 27), $j, k$ are matrix row- and column-indices (in our example, running from 0 to 789) for the exponentiated unitary matrix $U_P = \text{expm}(\cdots)$, $s$ is a style-index (here, running from 0 to $S - 1 = 78$), the term under

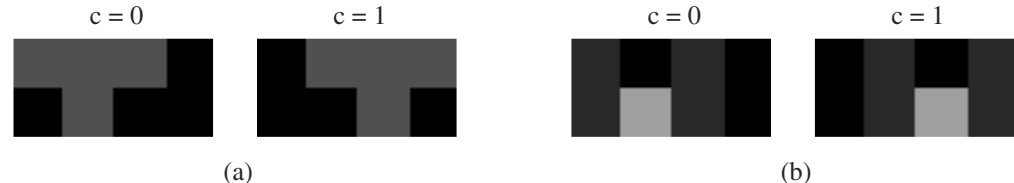

$$c = 0 \qquad c = 1 \qquad\qquad c = 0 \qquad c = 1$$

(a) (b)

Figure 1: (a) The two shapes of the toy example. The four gray pixels correspond to a photon arrival probability of $1/4$, i.e. a probability amplitude of $1/2$. (b) The per-pixel photon arrival probability after the orthogonal transformation is applied. The dark gray pixels correspond to a probability of $1/8$ and the light gray pixels to $1/2$.

the square root is the relative contribution of the $(y, x)$-pixel to the total brightness of example image $E$, and the $\delta$-factors are used for translating a pair of row,column image-indices to a linear pixel index, respectively an index on the $U_P$-rotated quantum state vector to a pair of (class, style)-indices. Technically speaking, from the viewpoint of mapping an optical amplitude that describes light intensity passing through the image-filter to the quantum amplitude of a particular (class, style)-combination, this is simply a linear model (since quantum mechanics is linear), whose linear weights are however specified in a slightly unusual way, underneath a complex matrix exponential (since quantum mechanics is unitary, i.e. probability-preserving). The probability to predict a given class $c$ is then obtained by summing over the probabilities associated with the given class (but different style-index).

Model accuracy has to be evaluated with caution: as we need to make a prediction after detection of a single photon, accuracy is the quantum probability of the correct label, averaged over all examples. Naturally, we can not determine which of the $C$ output classes would receive the most photons (= has highest probability) if all we have is a single photon detection event. This accuracy, about $40\%$ for the problems considered here, differs substantially from the accuracy that would be obtainable by looking at many photons coming from the same example image, which here typically exceeds $90\%$, roughly in alignment with the expected achievable performance of a linear model on MNIST. In other words, probabilities are uncalibrated, and the (non-linear "deep learning") transformation that would be required to calibrate them cannot be expressed as a unitary operator.

Let us consider a radically simplified example that illustrates why this method works. We want to discriminate between only two different shapes (with no further shape variation) on a $2 \times 4$ pixel screen where each pixel is either "on" or "off", using only one photon. Specifically, let us consider the two Tetris "T" shapes represented in figure 1(a).

For both shapes, the probability that the single photon arrives on one of the "on" pixels is $1/4$; therefore, taking into account that for two pixels the correct shape is identified exactly and for two with $50\%$ probability, we conclude that the baseline accuracy is $1/2 + 1/4 = 75\%$. Instead, we can apply a unitary transformation to reshape the probability amplitudes.

Let us now consider the simple but not optimal transformation of the photon amplitude that replaces the pair of amplitudes $(a, b)$ in each 2-pixel column with $((a - b)/\sqrt{2}, (a + b)/\sqrt{2})$, i.e. creates destructive interference in the top row and constructive interference in the bottom row. This gives the detection probability patterns shown in figure 1 (b). Maximum likelihood estimation here gives an accuracy of $1/2 + 3/8 = 87.5\%$.

Obtaining the maximum achievable accuracy will here require a more complicated all-pixel amplitude transformation, obtained as follows: The quantum amplitude transformation is angle-preserving, and the angle $\alpha$ between the two amplitude quantum states $q_1$, $q_2$ is given by $\cos \alpha = \langle q_1 | q_2 \rangle = 0.5$. Hence, we can rotate these two states to lie in the plane of the first two Cartesian coordinate axes of the Hilbert space, and at the same angle from their bisector. Identifying these coordinate axes with the correct labels, the accuracy is the cosine-squared of the angle between the transformed state and the corresponding axis, i.e. $\cos^2(\pi/4 - \alpha/2) = (\sqrt{3} + 2)/4 \approx 93.30\%$.

Table 1: Results for the Fashion-MNIST and MNIST datasets. The "classic" accuracy and information refer to the observation of a single photon, while the "quantum" quantities are obtained after applying the quantum transformation.

| Dataset | Entropy [bits] | Accuracy Bound (classic) | Information (classic) [bits] | Accuracy (quantum) | Information (quantum) [bits] |
|---|---|---|---|---|---|
| Fashion-MNIST | 3.32 | 21.38% | 1.10 | **36.14%** | **1.85** |
| MNIST | 3.32 | 22.96% | 1.20 | **41.27%** | **2.04** |

While the performance measure that we care about here is the probability for a correct classification, one observes that model training is nevertheless more effective when one instead minimizes cross-entropy, as one would when training a conventional machine learning model. Intuitively, this seems to make sense, as a gradient computed on cross-entropy loss is expected to transport more information about the particular way in which a classification is off than a gradient that is based only on maximizing the correct classification probability.

Overall, this task is somewhat unusual as a machine learning problem for three reasons: First, it involves complex intermediate quantities, and gradient backpropagation has to correctly handle the transitioning from real to complex derivatives where the loss function is the magnitude-squared of a complex quantity. TensorFlow is at the time of this writing the only widely used machine learning framework that can handle this aspect nicely. Appendix A.3 provides details on numerical aspects. Second, (as explained above), we cannot simply pick the class for which the predicted probability is highest as the predicted class. Rather, the probability for the single-photon measurement to produce the ground truth label sets the accuracy. Third, while most machine learning architectures roughly follow a logistic regression architecture and accumulate per-class evidence which gets mapped to a vector of per-class probabilities, we here have the probabilities as the more readily available data, so the computation of cross-entropy loss will have to infer logits from probabilities. Due to this need to compute logarithms of probabilities, it is very important that the training process does not intermediately see invalid probabilities outside the range $(0\ldots1)$, and this is ensured by parametrizing unitary transforms as matrix exponentials of anti-hermitean matrices.

TensorFlow code to both train such a model and also evaluate its performance is included in the supplementary material.

## 3 RESULTS

Figure 2 shows the most probable image class for each pixel, for the "Fashion-MNIST" and MNIST datasets. A classifier that looks up and predicts the most likely class in this table achieves maximal accuracy among all single photon classifiers that do not employ quantum interference. This includes classifiers that have had access to the test set during training. This accuracy is reported in the third column of table 1. (We note that, as pointed out by Sun et al. (2007), the "Fashion-MNIST" dataset contains many mislabeled instances, which affects both classical and quantum results.) We can compute the amount of information provided by the photon by computing the difference between the class entropy, i.e. $-\log_2(0.1) = 3.32$, since there are 10 classes, and the entropy associated to the classification errors, i.e. the accuracy. The mutual information for the classical classifier is given in the fourth column of table 1.

Training a unitary $U(790)$ quantum transformation that gets applied after the photon passed the image filter and before it hits a bank of 790 detectors allows boosting accuracy for both the "Fashion-MNIST" and MNIST datasets, as reported in the fifth column of table 1. The observation of the photon after the transformation provides a higher amount of mutual information with respect to the classical case. The values of mutual information in this case are given in the last column of table 1. Explicit matrices to perform the transformation for the two data sets have been made available with the supplementary material.

The quantum transformation $U_P$ allows us to define the pixel-space projection operators:

$$P_{\text{class } C} := U_P^{-1}\left(|\psi_C\rangle\langle\psi_C| \otimes I_{\text{style}}\right) U_P \qquad (9)$$

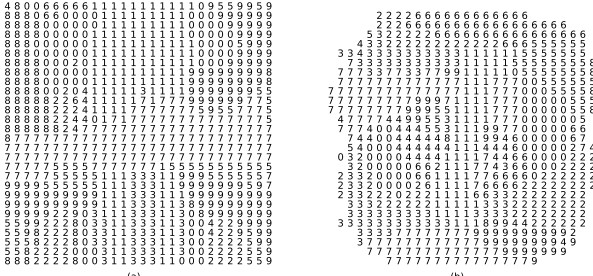

Figure 2: (a) Fashion-MNIST most likely class given detection of a single photon at the corresponding pixel coordinates. Here, the classes are: 0=T-shirt/top, 1=Trouser, 2=Pullover, 3=Dress, 4=Coat, 5=Sandal, 6=Shirt, 7=Sneaker, 8=Bag, 9=Ankle Boot. (b) Most likely digit-class given detection of a single photon for MNIST. A non-quantum classifier cannot outperform one that looks up its answer on the corresponding table.

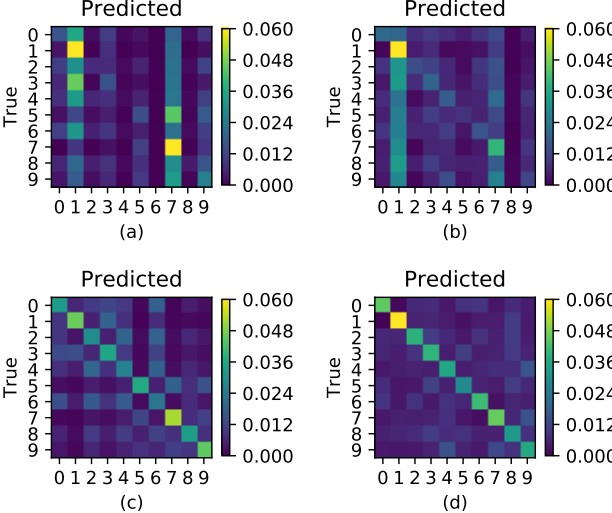

Figure 3: The confusion matrices for the "Fashion-MNIST" and MNIST datasets when classic and quantum classifiers are used: (a) "Fashion-MNIST"/classic, (b) MNIST/classic, (c) "Fashion-MNIST"/quantum, (d) MNIST/quantum.

with which we can decompose any example into contributions that are attributable to the different classes. Here, one must keep in mind that such separation is done at the level of probability amplitudes, so while we can compute intensities/probabilities from these components, which are mutually orthogonal as quantum states, summing these per-component per-pixel intensities will not reproduce the example's per-pixel intensities. This shows most clearly when considering the decomposition of an example "Trouser" from the "Fashion-MNIST" dataset's test set with the model we trained for this task, as shown in figure 4(a). The dark vertical line between the legs in the original image mostly comes from *destructive interference* between a bright line from the "Trousers" component and a matching bright line from the "Dress" component.

Due to the intrinsic quantum nature of this set-up, care must be taken when interpreting confusion matrices. Naturally, we never can claim of any single-photon classifier that it would 'classify a particular image example correctly', since re-running the experiment on the same example will not see the photon always being counted by the same detector! So, strictly speaking, for any single-photon classifier realized as a device, the "confusion matrix" could be determined experimentally only in the statistical sense, leaving uncertainty in the entries that decreases with the number of passes over the test set. Confusion matrices are shown in figure 3.

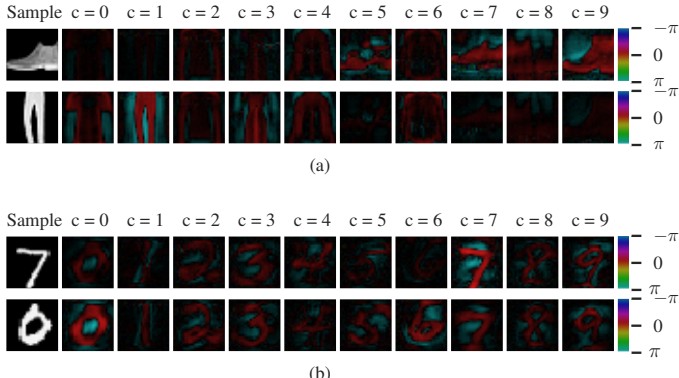

Figure 4: Projection of probability amplitudes for some samples of the "Fashion-MNIST" (a) and the MNIST (b) datasets. The first image shows the original sample probability and the following images show the probability amplitudes for each class. We visualize the complex amplitude by using brightness to represent magnitude and hue for phase (the colormap for the phase is shown on the right of each row.)

Our factorization ansatz appears to contain a hidden constraint: we are forcing each image class to use the same number of style-states. One could imagine, for instance, that a classifier might achieve even higher accuracy by treating image classes unevenly. Any such model that allows more style-space dimesions for some classes can always be embedded into a model that allows more style-space dimensions for all classes, so this question can be answered by padding to a larger Hilbert space. Numerical experiments, e.g. padding to 1000 rather than 790 dimensions, suggest that this has no appreciable impact on classification accuracy.

## 4 DISCUSSION

In summary, we demonstrated that, at least for the considered datasets, the space of the single observed photon state can be factorized remarkably well by a product of the example class space and a space collecting the remaining variables, such as the style. This factorization can be obtained easily by the proposed method and is experimentally realizable with optical elements placed in front of the sensor. The supplementary material contains a blueprint for an example circuit outperforming the classical limit (at $36.05\%$ accuracy) on $10 \times 10$ downsampled MNIST.

An experimental implementation of the proposed system would be a demonstration of a high-temperature and low- effective-qubit quantum ML device. With respect to other experimental approaches to quantum computing, such a device would have the limitation that it is built for the specific classification problem and cannot be reconfigured easily. It would be interesting to see whether an advanced quantum protocol along the lines of Knill et al. (2001) might enable the realization of more sophisticated intermediate-scale high temperature quantum machine learning in a way that mostly (like here) bypasses the need for quantum logic built from common quantum gates.

## 5 TENSORFLOW CODE

TensorFlow2 code to reproduce the experiments of this work and all the figures is provided in the ancillary files together with the computed unitary tranformations for MNIST and "Fashion-MNIST".

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

## A APPENDIX

### A.1 EXPERIMENT SCHEMATICS

Figure 5 shows the schematics of an experimental set-up: The lens (A) stylizes the last optical component of the monochromatic, coherent, linear-polarized (i.e. laser) light source that emits photons entering from the left and travelling to the right. Light intensity is controlled (e.g. by means of an absorbing filter, not shown) to be so low that photons travel through the apparatus individually. Any interference effects are hence due to self-interference of a single photon's wave function (just as in the double slit gedankenexperiment). The laser photons then hit a coherently illuminated $N \times N$ screen (B) (e.g. a LCD screen, in this diagram $10 \times 10$) which allows light to pass through a given pixel with coordinates $(y, x)$ with a probability that is proportional to the ink density on the example image. The photon arrives at each pixel with the same optical phase (i.e.

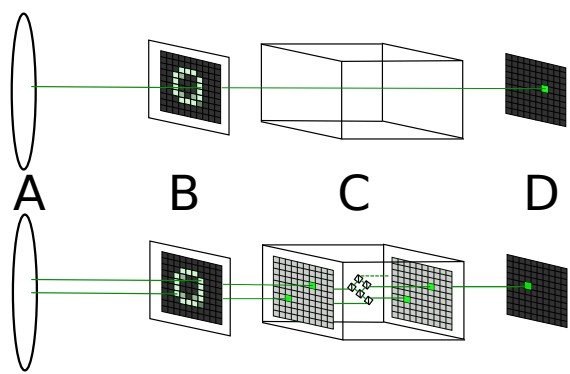

Figure 5: Schematics of the experimental set-up. Top: Classical Baseline, Bottom: Quantum Set-Up.

having travelled the same (fractional) number of wavelengths as seen from the laser). The diagram shows an exemplary raster image of a digit zero with maximal brightness (maximal ink density on the digitized ML example) on 14 pixels (with zero-based row/column coordinates $(y, x) = (2, 4), (2, 5), (2, 6), (3, 2), \ldots, (5, 7))$, with 75% brightness on three pixels (coordinates $(3, 3), (6, 3), (6, 5)$), and 50% brightness on one pixel (coordinates $(7, 6)$).

The 'Classical Baseline' set-up does not use interference and would work just as well in a world where photons are 'Particles of Light' that cannot self-interfere (as envisioned by Newton). The dimensions of the apparatus need to be such that, when the image-filter is brightly illuminated, it casts a sharp shadow on the detector-array. For the classical case, use of a coherent source of light is not necessary. At very low light levels, photons coming from the light source (A) will keep hitting the screen (B), frequently getting absorbed by a dark pixel. At some point, a photon will (by chance) manage to hit a non-dark pixel and not get absorbed (the more likely the brighter the specific pixel) and travel on to the single-photon detector array (D) (such as: a SPAD array) and be detected as having passed through a specific pixel. Ignoring experimental imperfections that could in principle be made small such as optical losses, the only possible transform on the photon state one could perform with a passive-linear optical device at (C) would be equivalent to coupling the photon into an array of optical fibers, routing each fiber from one pixel to some other pixel, and coupling out the photon at the other side of the device. This is equivalent to re-shuffling the pixels, which can always be un-done by re-shuffling the addresses of the cells of the detector array (D) and so does not affect classification accuracy – the diagram hence omits such transformations that cannot affect performance.

The quantum set-up (bottom) is perhaps easiest to analyze via Feynman's path integral interpretation of Quantum Mechanics: There are different 'histories' which lead from the same initial point (a photon coming from the laser) to the same final result (the photon being detected at a specific pixel, such as: 'at coordinates (4, 5)'), and the prescription is that we have to attribute a complex quantum amplitude to each such 'history', summing over all amplitudes that connect the same initial and final state to get the resultant amplitude, and obtaining the associated probability as the magnitude-square of the (complex) resultant probability. We can also consider the resultant per-pixel quantum amplitudes for a photon having travelled from the light source (A) not all the way to the detector but to some intermediate point, such as just after passing the screen (B). These are described by vectors with $N^2$ entries, one per pixel, whose absolute-magnitude-squares sum to 1. In the example, the photon-state-vector on any flat, screen-parallel surface between B and C, $|\psi_{BC}\rangle$, has zero entries for all but the 14+3+1=18 non-dark pixels. The entries $\psi_{BC}[24], \psi_{BC}[25], \ldots$ that correspond the 14 maximally-bright pixels at $(2, 4), (2, 5), \ldots$ are identical, as the photon reached each pixel at the same optical phase. Calling this amplitude $c_B$ (B for 'bright'), the amplitudes for the three moderately-bright pixels $c_M$, and the single dim-but-not-dark pixel's amplitude $c_D$, the amplitude magnitude-squares must be proportional to pixel brightness (probability for a photon to pass through the image) and sum to 1 (total probability for the photon that passed through the screen to have passed through a pixel). As the complex quantum amplitude phase matches the optical phase, these constraints fix $c_B = u \cdot \sqrt{1/Z} \approx 0.244$, $c_M = u \cdot \sqrt{0.75/Z} \approx 0.212$, $c_D = u \cdot \sqrt{0.5/Z} \approx 0.173$, with $Z$ being the overall normalization factor that makes the sum of magnitude-squares of all amplitudes 1, i.e. $Z = 14 \cdot 1 + 3 \cdot 0.75 + 1 \cdot 0.5$, and $u$ being some complex number of magnitude 1, the non-observable overall quantum phase factor. In the 'Quantum' set-up, we can employ a linear optical device, built from many beam-splitters and phase shifters (= 'delay lines'), to adjust self-interference of the photon wave function. The diagram shows two example paths out of the total $10 \times 10$ different paths that the photon can take before it reaches the detector array. Quantum Mechanics tells us that 'the single photon(!) travels along all these paths simultaneously' – this is just Young's double slit experiment in a slightly more complicated setting.

There is a 1-to-1 correspondence between physically realizable linear optical devices and probability-preserving (generalized) rotations of the quantum state vector. The (in this downsampled example: $100 \times 100$) components of such a transformation form a unitary matrix. If we used a linear optical device that implemented a random such transformation, and sent many photons through the apparatus, they collectively would produce an image conceptually resembling the interference pattern on photo film that codifies a hologram. Using a basic Machine Learning procedure, stochastic gradient descent, we can train the parameters of the transform such that photons coming from an image that shows a digit '0' preferentially land on the 0-th row of the detector array, photons coming from an image that shows a digit '4' preferentially land on the 4-th row, etc. The supplementary material describes a specific set-up in terms of optical components that reaches $> 36\%$ accuracy for MNIST downsampled to $10 \times 10$: If this were manufactured from ideal-quality optical components, the probability for a photon that passed through the screen (B) to land on the detector row matching its digit-class is better than $36\%$.

## A.2 CLASSICAL BASELINE ACCURACY THRESHOLD: MAXIMALITY PROOF

Elementary statistical considerations allow us to obtain a stringent upper bound for the maximum accuracy that cannot be exceeded by any classifier which satisfies these two properties:

- P1: The classifier must make a prediction using as its input only the index of the one detector in the detector-array that received the first photon. It can not use any additional information about the

example (but may have been trained *with arbitrary information about the example set*, even including full knowledge of the training *and* test set).

- P2: There is a one-to-one correspondence between image pixels and detector-cells: For each image-pixel, there is exactly one detector-cell such that when the example image $E$ is presented, then the probability for the first photon to land on the detector cell $k$ is proportional to the brightness of the associated pixel in image $E$.

This bound is what we call the 'classical accuracy bound'. The 'quantum' classifier violates P2 by employing photon self-interference: The probability for the $k$-th detector to observe the first photon depends on collective information which the photon 'holographically' transports about the input image once it passed the image-filter, rather than on a single pixel.

The protocol for evaluating classifier accuracy is as follows: *We pick a random example from the dataset's test set, set up the device to present this example as a problem, send light towards the filter-screen, and look at the first photon that managed to pass the filter-screen and get counted by the detector array, then map the index of the detector that counted the photon to a predicted class. We register 'successful prediction' if the predicted class matches the example's label, otherwise we register an 'unsuccessful prediction'. The accuracy is the probability of the prediction to be successful.*

Somewhat unusually, this means that there is no such thing as 'the predicted class of a given image', as one normally would have it in a Machine Learning problem. This is due to the inherent randomness of quantum mechanics: Even repeating classification for the same image multiple times, we will see photons land on detectors that correspond to different classifications. If we were to experimentally determine accuracy, this would then suffer from the usual problems of determining a probability via a statistical experiments: one can make the likelihood to be way off arbitrarily small, but never shrink it to zero. However, the aforementioned protocol makes it possible to directly compute the maximal achievable probability for any classifier, without resorting to a statistical experiment.

The gist of our argument parallels the reasoning behind the claim that we can show with simple statistics that no ML classifier can possibly outperform an accuracy threshold of $29/36$ for predicting from the eye total of rolling two dice whether any of the two dice showed a six. Here, the reason is that we can get maximal accuracy by looking look at all the possible realizations of any given eye total and make the best possible guess given the situation. For eye totals $2 - 6$ $(1 + 2 + 3 + 4 + 5 = 15$ of 36 cases), we would predict 'No' and always be correct. For eye totals 11 and 12 $(2 + 1$ additional cases), we would predict 'Yes' and also always be correct. For each other eye total, there are two realizations where one die shows a 'six', so we would want to predict 'Yes' for eye totals having at most four realizations, i.e. where we have at least a $50\%$ chance of being correct, and 'No' otherwise. Using this approach, we would incorrectly classify three cases as 'Yes' $(5 + 5, 4 + 5, 5 + 4)$, and incorrectly classify four cases as 'No' $(6 + 1, 1 + 6, 6 + 2, 2 + 6)$. Except for these $7/36$ cases, we would make a correct prediction, so optimal accuracy is $29/36$. Given the perhaps somewhat unfamiliar 'quantum ML' setting, and the need to rigorously justify the optimality claim, we prove it below. The only material difference to the dice-sum example is that relative weights of realizations are not determined by counting, but by looking at pixel brightnesses.

In analogy to the the dice example, the key observation is that the classifier's input is a single pixel-index, and its output is an image-class. So, we can completely specify any (deterministic or not) classifier's action by tabulating, per-pixel-index, what the probability is for this classifier to map the given input pixel-index $k$ to each of the possible output classes $c$. The resulting matrix $K_{kc}$ would, for a deterministic classifier, simply be a matrix with one-hot encoded image class, one row per pixel-index. The classifier's accuracy is then given by

$$
\begin{aligned}
\text{Accuracy} \quad &= \quad P(\text{Classification is correct}) = \sum_E P(E) \sum_k \cdot P(\gamma_k|E) \cdot P(y_E = C(\gamma_k)) = \\
&= \quad \sum_{E,k} P(E) \cdot P(\gamma_k|E) \cdot K_{k,c=y_E}.
\end{aligned}
\tag{10}
$$

Here, $P(E)$ is the probability to pick example $E$ from the test set (i.e. $1/\{\text{test set size}\}$), $P(\gamma_k|E)$ is the probability to detect the photon in the detector cell with index $k$, given the example $E$, and $P(y_E = C(\gamma_k))$ is the probability that example $E$'s label $y_E$ matches the classifier's output on the input "the photon was detected in cell $k$".

The probability for detecting a photon in cell $k$ when randomly drawing an example image from the test set is $P(\gamma_k) = \sum_E P(E) \cdot P(\gamma_k|E)$. The probability for a fairly drawn example's label to be $y_c$ when a photon was detected at cell $k$ is $P(y_c|\gamma_k) = \sum_E P(E) \cdot P(y_E = y_c) \cdot P(\gamma_k|E)$. Let us tabulate these $P(y_c|\gamma_k)$ in the $\{\#\text{pixels}\} \times \{\#\text{classes}\}$ matrix $R_{kc} := P(y_c|\gamma_k)$.

We then have:

$$
\text{Accuracy} = \sum_{\text{Detector cell } k} \sum_{\text{Class } c} P(\gamma_k) \cdot P(y_c|\gamma_k) \cdot K_{kc} = \sum_{k,c} P(\gamma_k) R_{kc} K_{kc}.
\tag{11}
$$

In words: We can compute accuracy by looking at each detector cell $k$ and each class $c$, determining the probability $P(\gamma_k)$ that, when fairly drawing examples from the test set, a photon gets detected at cell $k$, and

splitting up this probability into contributions from examples where the target class was 0, 1, 2, etc. These contributions are $P(\gamma_k) \cdot P(y_c|\gamma_k)$. We make a correct classification when the classifier also predicts class $c$ given the input $k$. The classifier's behavior when given the input $k$ is specified by row $k$ of the $K$-matrix, so this probability is $K_{kc}$.

Here, $P(\gamma_k)$ and $R_{kc}$ are determined by the test set. Each admissible matrix $K_{kc}$ that has $\sum_c K_{kc} = 1$ specifies a different classifier, and the accuracy is a function of this matrix $K$ only. The question is now which admissible matrix $K$ maximizes accuracy. Total classification performance (accuracy) is a weighted sum over per-detector-cell performances (the weights being the probabilities to observe a photon in cell $k$ when doing detection experiments on samples drawn fairly from the test set). Let $K_1$ be a matrix that maximizes accuracy, and $K_2$ be a matrix obtained by picking, for each cell-index $k$, a probability row-vector that maximizes $\sum_c R_{kc} K_{2,kc}$. We have Accuracy($K_1$) $\geq$ Accuracy($K_2$) (since $K_1$ is optimal), and also $\sum_c R_{kc}(K_{2,kc} - K_{1,kc}) \geq 0$ (since $K_2$ maximizes this value on each row $k$), so, taking a weighted sum with weights $P(\gamma_k) \geq 0$, we find $\sum_k P(\gamma_k) \sum_c R_{kc}(K_{2,kc} - K_{1,kc}) \geq 0$, i.e. Accuracy($K_2$) $\geq$ Accuracy($K_1$), hence Accuracy($K_1$) = Accuracy($K_2$). In words, we achieve maximal accuracy if we individually look at each "photon detected in cell $k$" case and make the optimal prediction there. Now, for a fixed cell-index $k$, $\sum_c R_{kc} K_{2,kc}$ is maximal if the matrix-row $K_{2,kc}$ has an entry 1 for the index $c$ for which $R_{kc}$ is maximal, and is zero otherwise. To see this, let us assume $K_{2,kc} > 0$ for some index $c$ for which there is another class index $d$ with $R_{kd} > R_{kc}$. Then, incrementing $K_{2,kd}$ by $K_{2,kc}$ and subsequently setting $K_{2,kc}$ to zero increases $\sum_c R_{kc} K_{2,kc}$, which is a contradiction. So, optimal choices of $K_{2,kc}$ are zero for classes $c$ for which $R_{kc}$ is not maximal. Also, we always attain the maximum when choosing each row-vector of $K$ to be one-hot and have its 1-entry in a place that maximizes $R_{kc}$, i.e. maximal achievable accuracy is obtained by a classifier which, for every cell-index $k$, predicts the most likely digit-class subject to the constraint that a randomly drawn example from the test set had its first photon-detection occur at detector cell $k$. This theoretical upper bound on classifier accuracy hence is given by:

$$\text{Accuracy} \leq \sum_k P(\gamma_k) \max_c R_{kc}. \tag{12}$$

For the MNIST dataset, this is found to be 22.957% (rounded up to 22.96%), while for Fashion-"MNIST", we get 21.375% (rounded up to 21.38%). Code that implements this calculation is available in the supplementary material. We should emphasize that the constructive procedure described here that yields a classifier attaining this stringent upper bound does inspect the test set, and relevant deviations in statistical properties between training and test set would manifest in the form of lowering attainable accuracy for a classifier that is trained on the training set only.

### A.3 BACKPROPAGATION WITH COMPLEX INTERMEDIATE QUANTITIES

As explained in the main text, the per-class probabilities defined by Eq. (8), when used as input to a conventional softmax loss function, make training the real weight-parameters matrix $W_P$ in terms of which the unitary rotation is expressed a straightforward procedure.

Nevertheless, this approach utilizes some capabilities which at the time of this writing are likely TensorFlow-specific. It hence may make sense to describe the training procedure in sufficient detail to allow straightforward re-implementation on top of some other Machine Learning framework, or perhaps even directly without use of any such library.

The loss function is defined in terms of the magnitude-squared of a complex intermediate quantity, which here is the vector of complex quantum amplitudes, one entry per class/style combination. In this appendix, we henceforth consider the simplified $10 \times 10$ problem described in detail in appendix A.1.

We can perform the calculation entirely in terms of real quantities by replacing every complex number $C + iD$ by a real $2 \times 2$ matrix block of the form

$$C + iD \rightarrow \begin{pmatrix} C & -D \\ D & C \end{pmatrix}. \tag{13}$$

This means in particular that a 100-dimensional (complex) amplitude-vector $a_j$ gets replaced by a $200 \times 2$-matrix $A_{mn}$. If we interpret the $a$-index $j$ as encoding *class* $c$ and *style* $s$, i.e. $j = c \cdot S + s$, the total probability for class $c$ is $p(c) = \sum_s |a_{c \cdot S+s}|^2 = (\text{Re } a_{c \cdot S+s})^2 + (\text{Im } a_{c \cdot S+s})^2$, and this gets replaced by $p(c) = \sum_s \left( A_{(c \cdot S+s) \cdot 2,0}^2 + A_{(c \cdot S+s) \cdot 2,1}^2 \right)$ (reading off the real and imaginary part from the 1st column of the $2 \times 2$ block that represents $a_j$).

As the square root of the relative per-pixel intensity is real, the input-image amplitudes in this approach likewise get represented by a $200 \times 2$-matrix $B$. Specifically, if e.g. $Q_{2,5}$ is the contribution of pixel $(y = 2, x = 5)$'s

brightness to the total image-brightness, this gets represented as:

$$\begin{pmatrix} B_{(2\cdot10+5)\cdot2,0} & B_{(2\cdot10+5)\cdot2+1,0} \\ B_{(2\cdot10+5)\cdot2,1} & B_{(2\cdot10+5)\cdot2+1,1} \end{pmatrix} = \begin{pmatrix} B_{50,0} & B_{51,0} \\ B_{50,1} & B_{51,1} \end{pmatrix} = \begin{pmatrix} \sqrt{Q_{2,5}} & 0 \\ 0 & \sqrt{Q_{2,5}} \end{pmatrix}. \quad (14)$$

The off-diagonal part, which would correspond to the imaginary part of the amplitude, is zero here. The matrix that gets exponentiated is a real $200 \times 200$ matrix, and its exponential, which also is a real $200 \times 200$ matrix, gets multiplied from the right with the $200 \times 2$ matrix of input-image amplitudes and gives the real $200 \times 2$-matrix $A$ from above that contains the real and imaginary parts of class- and style-amplitudes.

The real $200 \times 200$ matrix under the exponential only depends on $100 \times 100$ real parameters $W_P$. Calling the $200 \times 200$-matrix $M$, the "$2 \times 2$-blocking" prescription to obtain its entries from $W_P$ is:

$$\begin{pmatrix} M_{i\cdot2\ \ ,j\cdot2} & M_{i\cdot2\ \ ,j\cdot2+1} \\ M_{i\cdot2+1,j\cdot2} & M_{i\cdot2+1,j\cdot2+1} \end{pmatrix} = \begin{pmatrix} (W_{ij} - W_{ji}) & -(W_{ij} + W_{ji}) \\ (W_{ij} + W_{ji}) & (W_{ij} - W_{ji}) \end{pmatrix}. \quad (15)$$

Finally, we need a backpropagation-friendly prescription for computing a good approximation to the matrix exponential. The theory of compact Lie groups tells us that we can reach every 'generalized' (since complex) rotation matrix by exponentiating matrices where each entry is from some not too large interval. For matrices with small entries only, we can use a truncated Taylor polynomial to get a good numerical approximation of its exponential, using

$$\mathrm{expm}(M) \approx I + M + \frac{1}{2}M \cdot M + \frac{1}{6}M \cdot M \cdot M + \ldots, \quad (16)$$

and we can reduce the problem of finding the matrix exponential of a matrix where this series requires many terms to give a good approximation by repeated halving and squaring, repeatedly using the property $\mathrm{expm}(M) = \mathrm{expm}(M/2)^2 = \mathrm{expm}(M/2) \cdot \mathrm{expm}(M/2)$.

For the problem discussed here, the angle-ranges for rotations that need to be considered are limited, and this makes it feasible to in-advance pick both a number of squarings (such as: 8) and a maximal term in the Taylor expansion (such as: 10th power), and get very good results.

The numerical computation implemented in the supplementary material, being based on TensorFlow, deviates from the procedure described here in two relevant ways. First, while TensorFlow's differentiable matrix exponentiation algorithm employs repeated halving/squaring, it uses a Padé rather than Taylor approximation to compute the matrix exponential of a matrix with small entries. Second, TensorFlow can directly backpropagate through complex intermediate quantities, and handles the transition between real and complex gradients in just the way that one also obtains when expanding complex numbers to real $2 \times 2$ blocks as described above. It can however avoid the inefficiency associated with using actual real $2 \times 2$ matrix blocks that make every real and every imaginary part show up in memory not once, but twice.

## B  CHANGES

For the sake of transparency and providing a better overview over the evolution of this document, we added this appendix that tracks changes made during the review period. This appendix will be removed at the end of the review period.

### B.1  2020-11-13 CORRECTION→ 2020-11-23 UPDATE

- Made the changelog a separate appendix.
- Fixed an explanation of the baseline-accuracy in the main text: This still referred to proven-optimal-on-examples-generated-from-training-set, even as the updated baseline now already referred to classifiers that even allow 'cheating', in the form of training on the test set.
- Slightly adjusted the wording in appendix A.1 to clarify that apparatus dimensions are such that interference effects are negligible when no linear optical device transforms the photon-state and the image-filter casts its shadow on the detector-array.
- Figure 2: Added explanation to the caption that these figures actually describe a maximal-accuracy classifier.
- Added another appendix on complex-backpropagation as described in our 2020-11-19 reviewer-response.

### B.2  2020-11-13 RESPONSE→ 2020-11-13 CORRECTION

Formula (8) required a fix: The sum over style-parameters is, in quantum mechanical language, an 'incoherent summation' rather than 'coherent summation': After the unitary transform, as the photon hits a detector, this gives

us information about the class (for the downsampled $10 \times 10$ problem discussed in appendix A.1, via the row of the detector-cell), and also the style-parameter (via the column of the detector-cell). After this measurement, we know 'class' *and style*, but decide to ignore 'style'. Still, as the 'style' is the outcome of a measurement, there can be no inference between different 'style' results, and so we need to sum the probability-contributions for all the different style-outcomes, rather than summing amplitudes. We apologize for this rather obvious oversight that happened as we somewhat rushed our first response in order to give reviewers an early answer.

### B.3 Initial Submission→ 2020-11-13 response

Major changes were:

1. Re-structured introduction, introducing a 'related work' section and expanding in the main text the explanation of the relation of this work to other recent analog all-optical image classification research.

2. Added formula (8) that directly expresses the probability for a given example to be classified in a particular way in terms of image pixel data and model parameters.

3. Replaced the classical baseline accuracies (which were for any classifier that was trained with access to the training set only) with strict proven accuracy bounds *that can not even be beaten by classifiers that were trained with knowledge of the test set*. So, these hard thresholds for non-quantum classifiers now can not be outperformed by any means, *including* 'cheating' in the form of training on the test set. The impact is that the baseline performance threshold rises from $21.28\%$ to $22.96\%$ (MNIST) and from $18.28\%$ to $21.38\%$ (F-MNIST), requiring adjustment of some plots.

4. Added a self-contained appendix with schematics alongside a detailed explanation of how image information gets translated to a photon quantum state.

5. Added a self-contained appendix with the detailed proof (including a proof idea sketch) of the claimed maximum attainable classical accuracy bound.

