# OpenReview forum: "Single-Photon Image Classification"
_ICLR.cc/2021/Conference — ICLR 2021 Poster_

### Official Review · AnonReviewer2 · 2020-10-23
**Interesting work**

**Rating:** 7
**Confidence:** 3

**Review:**

The goals of this work are ambitious: to clearly define a setting useful to both physicists and machine learning practitioners.

The current paper is an excellent step in this direction.

However, with several semesters of undergrad quantum mechanics courses, I found it difficult to follow the calculations needed to compute predictions.

Moreover, given my graduate-level experience in machine learning, it was even more difficult to clearly understand the data and training algorithm.

In the current state I do not think ICLR is an appropriate venue for this work, as it may confuse machine learning practitioners. (Though, I think the original goal of the paper is very worthy, and look forward to the final version of this work!)

To try to give constructive critique:

- it took me a long time to understand the setup of the problem. An illustration of FashionMNIST or MNIST would help, with an incoming photon, an arrow through the cut-out of the image (clearly delineating that the image is binarized, and that white pixels are cut out, allowing the photon to pass through), and a detector (An LCD screen is described — is this the detector as well? Or does the photon bounce back and is then detected?). Such an illustration would go a long way for a machine learning practitioner unfamiliar with the double slit experiment and whatnot.

- an algorithm box. A machine learning audience is used to thinking in terms of training data and algorithms. In this case, an algorithm box would help, specifically as it shows where the additional information is coming from. It seems like the 'training' is in optimizing the parameters of the unitary transform. Clearly delineating input, output, and optimization steps will help clarify the method.

- an equation for computing the predictions of the trained model. Given a photon that passed through a mask, a clear formula for computing the class probability with the trained unitary operator parameters.

- clearly describing the baseline. It was hard to find the details of the classical performance reported. 'maximal classical performance' is confusing wording, and implies that 'maximal' is proven theoretically. Is there a citation for this? I may have missed it. If it is not proven theoretically, then the wording should be changed, and a clear description of the architecture, training data, and training algorithm should be used, and code should be included in the supplement. This will help machine learning folks understand exactly what the comparison is against. From a machine learning standpoint, is the single pixel input to the model randomly sampled every time?

- developing an additional baseline. Finding a unitary transform to find bases corresponding to style and class is unfair to the classical method, which does not have access to this information. Not having a baseline that uses this additional information will further confuse machine learning folks, as it seems obvious that a model that uses additional information will outperform a 'classical' model that does not use this information. For example, a tensor decomposition/SVD that uses information about style and class might be possible.

Hope this is helpful; I think with this additional work it could be quite a valuable contribution, as I think the ICLR community could be inspired to develop more methods that require complex-valued numbers such as this one.

Edit:

- the authors meaningfully addressed the above points. I have raised my score accordingly.

---

> ### Author Response · Authors · 2020-11-13
> **Substantially expanded supplementary explanations, added summarizing formula to main text.**
>
> We would like to especially thank this reviewer for clear and actionable feedback on how to make this work more accessible to an audience of readers with very inhomogeneous and perhaps often lacking background on quantum mechanics. This is extremely valuable feedback to us. Given that we have to expect very uneven distribution of prior knowledge of this subject in the readership, it might make sense to provide additional explanations that go beyond the core content of the paper in appendices. The major changes we made to the first revision of our manuscript are:
>
> 1. We slightly restructured the introduction and expanded the discussion how our work
>   is related to other recent research on optical ML.
> 2. We added formula (8), alongside a detailed explanation.
>   This compact formula shows explicitly how the probability for each output class is computed
>   from the input image, given the (trained) model parameters.
>
> Unfortunately, with these two additions, we mostly reached the extended 9-page limit. So, for now, we moved the more detailed explanations into two appendices. We are looking forward to working with our reviewers in order to eliminate some redundancy from now having both a compact and a detailed explanation of some aspects, and ultimately perhaps folding some of the rather relevant additional material - such as the schematic diagram - back into the main text of our article. Concerning the new appendices:
>
> 3. We are providing a schematic diagram in appendix A.1, alongside a very detailed explanation that specifically shows how the structure of the image gets encoded by the photon quantum state, clarifying in particular that there is no binarization.
>
> 4. We also provide a detailed proof of optimality for the "classical" (i.e. no-self-interference) baseline in appendix A.2, both explaining the basic idea by means of a simplified example, and also showing all the mathematical steps that lead to the conclusion that no classifier that does not use interference can possibly have an accuracy higher than this threshold.
>
> Checking our original submission's baseline, we found and fixed two implicit assumptions that one could regard as questionable. For the resubmitted manuscript, our new claims are (a) that the - slightly higher - "classical baseline" performance now can not even be beaten by any classifier that has properties P1 and P2, even if this might have had access to the test set during training, and that (b) the reasoning given in section A.2 constitutes a stringent mathematical proof of this fact.
>
> These changes directly address item 1 (diagram), 3 (complete formula with explanation), and 4 (precise description of the baseline) from the reviewer's list of improvement suggestions.
>
> Concerning the final point on the list of improvement suggestions, providing an additional baseline: We would like to address this in a separate comment.

---

> ### Author Response · Authors · 2020-11-13
> **On: Additional Classical Baseline - also: Algorithm Boxes**
>
> On the question of providing an additional baseline: We see an important aspect here that needs to be clarified first. Given the optimality proof in appendix A.2, we know that we can not get above our classical accuracy threshold unless we break property P1 or P2. This means in particular that we disagree with the idea that "it seems obvious that a model that uses additional [style and class] information will outperform a 'classical' model that does not use this information": Our proof in particular also means that a classical model that was trained not only having access to the test set, but also access to any other statistics of any example dataset (irrespective of training or test), such as in particular the suggested tensor decomposition / SVD (i.e. PCA) can not beat this threshold.
>
> We do agree that observing such a large large performance gap should generally be met with skepticism and seems to be asking for an explanation.
>
> On the "classical baseline" side, a perhaps useful question to ask is: if all we see is a single pixel flashing up once, do we really believe we can predict the digit class it came from with an accuracy that is far better than random guessing? Even 25% accuracy should be regarded as high in that light. On the quantum side, we have a witness that shows that better-than-40% accuracy is attainable - so, what explains that gap? The sole explanation for this gap is the ability to exploit interference of the quantum state of a single photon with itself - precisely what Figure 1 and the accompanying text explain in a simplified setting. With "Schrodinger's Cat", it is questionable whether the gedankenexperiment actually tells us anything about quantum mechanics that would need a deep explanation. Here, however, there is a clear, large, quantitative performance gap between what we can at best hope to achieve classically and what has been demonstrated to be possible with quantum mechanics: Our paper makes the often discussed "inherent spookiness" of quantum mechanics visible in yet another way - here, in a ML setting via its implications for an image classification task.
>
> Finally, and related to the above point, considering item 2 on the reviewer's list, algorithm boxes, especially for training: We did not yet address this in our first revision, in the interest of iterating fast and getting as much of an improvement out of our the review process of this article as we can (with all the reviewers' help, for which we are very grateful here).
>
> The supplementary material already contains a code implementation of the training procedure, which simply amounts to optimizing cross-entropy loss for predictions obtained with formula (8) via stochastic gradient descent on large batches of example images. (Large batches are useful here in order to amortize the cost of exponentiating a complex 790x790 matrix.) On the technical side, there is slightly more to this procedure than meets the eye, since we are actually relying on TensorFlow using a somewhat unconventional definition of a complex gradient, which happens to be not a proper complex derivative, but exactly the variant that one needs when backpropagating from a real amplitude-magnitude-square result through the contributing complex amplitudes, as in this example. One way to resolve this subtle technical detail would be to simply present the complex matrix calculation in the form of a real calculation with twice as large matrices and vectors, i.e. describe an implementation that would be half as efficient as the one actually provided by the supplementary material. The alternative would be to describe in detail TensorFlow's strategy for handling complex gradients, which unfortunately would however require introducing some complex calculus in addition to the already present need for proceeding in small steps when explaining quantum mechanics. We are happy to add another appendix that explains this using either option, but would ourselves generally lean towards avoiding any discussion of complex-valued backpropagation in this work. Feedback on what readers might likely find more accessible here would be highly appreciated.

---

> ### Author Response · Authors · 2020-11-19
> **Kindly asking for further clarification**
>
> Considering that the ICLR "author/reviewer discussion period" ends on 2020-11-24 (i.e.: coming Tuesday), and there are open questions about whether the revised version we submitted on 2020-11-14 indeed appropriately addresses the reviewer's concerns, we would like to ask the reviewer about their thoughts on these currently unresolved points:
>
> 1. Do they agree that the new appendices A.1 and A.2 appropriately address the need for more detailed explanations they flagged up?
>
> 2. Do they agree that formula (8) describes the computational aspects sufficiently well to also answer the questions that an "algorithm box" on the inference process would answer? This is, after all, basically a linear model (since quantum mechanics is linear), with some minor subtleties due to occurrence of complex intermediate quantities and unusual parametrization.
>
> 3. Do they agree with our proof of the accuracy threshold for the classical baseline? To re-state the claim: unless one uses quantum interference, maximal performance on the test set is obtained by taking the (x,y) pixel coordinates of the photon, looking up the corresponding entry in Figure 2, and predicting the class from that cell. No non-interference classifier will be able to achieve a higher accuracy on the test set than this procedure. (If they agree, this should remove the need for another classical baseline, which they asked for in the final two items of their initial response - we know the maximally attainable performance, "even when cheating is permitted".)
>
> 4. (As per our last response): What is their guidance for whether an explanation of the training process should explicitly discuss the subtleties around backpropagating through complex intermediate quantities. Should this be discussed in terms of a language that expresses everything as real quantities, or would it be appropriate to discuss this in terms of holomorphic and antiholomorphic functions, i.e. Wirtinger calculus?
>
> If we do not hear back before the end of the weekend, our plan is to assume that the answer to (1, 2) is that their concerns are appropriately addressed, that they (3) agree on our baseline accuracy threshold proof, and there is no need to explain this further, and we will address (4) by adding another appendix that explains complex-backpropagation in pedestrian terms by mapping it to real-backpropagation with twice-as-large matrices, which would work, but be somewhat less efficient than what TensorFlow actually does in such a situation.

---

> ### Comment · AnonReviewer2 · 2020-11-23
> **Response to authors**
>
> I thank the authors for a detailed response, and would like to see this paper accepted. I think it will spur a meaningful discussion at ICLR and lead to further work in this area.
>
> The authors' changes in the main text and appendix make the paper easier to understand.
>
> However, I encourage the authors that if the paper is accepted (or a revision is necessary) to include a concise explanation of where the additional information is coming from (in terms of mathematics, or an algorithm box, or ideally both to build intuition).

---

> > ### Author Response · Authors · 2020-11-23
> > **Thanks for clarification.**
> >
> > We thank the reviewer for taking another look at our work and clarifying that our changes do address their original concerns. We polished a few details and added another appendix that describes key aspects of the gradient calculation for training in more detail, as we realized that this information is useful when trying to independently reproduce our results without TensorFlow - where support for complex intermediate quantities may well be entirely absent. The changelog (now appendix B) shows these changes.
> >
> > With respect to where the extra information is coming from, the example in Figure 1 tries to clarify this question. We agree that the underlying mechanism is rather non-obvious and takes a while to understand. (We did not find this example straightaway, either.) Overall, the "quantum mechanical miracle" that we exploit in our work nowadays also has remarkable applications such as neutrino detectors weighing only a few kilograms(!) rather than many tons - see e.g. https://www.sciencemag.org/news/2017/08/milk-jug-sized-detector-captures-neutrinos-whole-new-way
> >
> > Despite limited time remaining, we are now focusing on finding ways to clarify the mechanism even better. The one problem that is just very hard to get around is that self-interference totally runs counter to human intuition.

---

### Official Review · AnonReviewer3 · 2020-10-28
**A new approach for quantum computing based machine learning**

**Rating:** 6
**Confidence:** 3

**Review:**

This paper focuses on the quantum computing based machine learning and proposes a toy model to illustrate the quantum information processing. On the common used handwritten digit dataset MNIST, more than 40% images can be classified accurately. The proposed method looks interesting and the focused problem (combining quantum computing and machine learning) is of certain significance.

Strength:
+ The topic is interesting, which inspires the following researchers to focus on the combination of quantum computing and machine learning. Both the theoretical analysis and experimental results demonstrate that the proposed classifier works well.
+ The visualization results in Figure 4 are interesting. With the proposed photon classifier, the semantic information of the input images can be well extracted. The photon classifier tends to produce large amplitudes for the right classes.
+ The proposed method is analyzed in detail from multiple perspectives, including results on MNIST, confusion matrices and visualization.

Weakness:
+ In the Results section, only experiment results of the proposed method is shown. There are some previous works related to this paper, such as [r1]. More comparisons and discusses between the proposed method and previous methods are desired.
+ The experiments are conducted on two simple datasets, MNIST and Fashion-MNIST. Real image data are more complex, such as colored natural images. Could the proposed method be applied on more complex data? If can, how to extend and apply it? Please provide such a discussion.
[r1] Erfan Khoram, Ang Chen, Dianjing Liu, et.al. Nanophotonic media for artificial neural inference. Photon. Res., 7(8):823–827, Aug 2019.

---

> ### Author Response · Authors · 2020-11-13
> **Expanded research context discussion in main text, clarifying generalizations to more complex data in response.**
>
> We thank the reviewer for their expert assessment, and agree that a clearer explanation of the relation between [r1] and our work is beneficial. We updated the draft to include a more in-depth explanation in the main text. Extending this particular approach to colored images would in principle be feasible, but introduce complications on the experimental side without adding interesting new aspects to the mathematical analysis of this problem: One would have to use a coherent light source that can produce collimated photons not at one, but three different wavelengths (colors), then either spatially separate the color-components or optically couple such photons into and out of the device that performs shaped self-interference in a way that simultaneously minimizes losses for different wavelengths, optimize the linear optical device parameters to simultaneously transform quantum states in a useful way for different input photon wavelengths, and finally also discriminate photons at different wavelengths. Considering that even when generating a color image, the widely employed technical solutions all approximate color by splitting a pixel into spatially separated sub-pixels, the most viable experimental approach might be to spatially separate the quantum state space into a red/green/blue component as early as possible (perhaps using mirrors that are transparent at different wavelengths, or photonic crystals) and use three individually trained instances of a linear optical amplitude-reshuffling device plus detector array. Likewise, going to higher resolution / more complex scenes would mostly introduce complications on the side of an experimental realization.
>
> Having said this, our general impression is that representing high level concepts that are relevant for other Machine Learning problems in terms of rather simple quantum states indeed might be promising to try. If we may wildly speculate on this, then machine learning tasks involving not images but graph structured data might turn out to become particularly interesting applications of such "analog quantum computing" architectures.

---

### Official Review · AnonReviewer1 · 2020-10-29
**Bringing the ML and QC worlds together using single-photon classification**

**Rating:** 3
**Confidence:** 1

**Review:**

The paper studies that  a ML system using quantum interference gives better classification accuracy than a vanilla ML system, under the  constraint that a classification decision has to be made after detection of the very first photon that passed through an image-filter.

The reader can gather that this work brings together the ML and QC worlds but it is not clear what the real motivation of this work is and primarily why is ‘single-photon’ important. Does single photon equate to a single pixel? Or is this denoting the very first photon that passed the filter? Also is this constraint of detecting the very first photon valid?

It might be good to know which audience reads the paper. If it is the ML audience, then a section on QC basics/terminology will help. Or at least a graphical abstract to drive home the point home, will be helpful.

Was there a reason to use the Fashion-MNIST in conjunction with MNIST dataset? The authors can also consider to abbreviate Fashion-MNIST to F-MNIST throughout the paper.

---

> ### Author Response · Authors · 2020-11-13
> **Added more explanations to paper and its appendices, clarifying questions about this work in the response.**
>
> We thank the reviewer for their assessment of our work, even if currently only labeled an "educated guess" and hope that the added material in the first revision makes it more accessible. On the question what the "real motivation" of this work is, one way to answer it is to see this article as a demonstration that high-level knowledge representation of concepts as abstract as "image of a shoe" can in principle be done remarkably well even using the most low-level concept that physics allows, namely the quantum state of a single photon. The significance of this is that there may be interesting "analog quantum computer" realizations of quantum ML that are not based on far-future "digital error-correcting quantum computer" architectures: We might not have to wait for a general-purpose many-qubits quantum computer for quantum ML to become practically relevant - very simple quantum systems can already encode high level features. On the physics side, multiphoton (not many-photon) quantum optics and potential Machine Learning applications certainly is an interesting topic to discuss, but would require substantially heavier mathematical machinery than what we felt comfortable showing to a general ML audience. Detecting single photons with spatial resolution is routinely done in quantum optics research, even at room temperature (typically via SPAD arrays), and we indeed consider it feasible to experimentally build a simplified version of the basic design presented in this work.
>
> The reviewer's questions whether 'single photon corresponds to single pixel' (Yes for the "classical baseline" that can not use self-interference of a photon, No for the quantum case), and whether "single photon denoted the very first photon that passed the filter" (Yes for both the "classical baseline" and "quantum" case, as explained already in the abstract) were, in our view, already addressed by our original submission, but we agree that this work clearly would benefit from a schematic explanation of our set-up. We added an appendix with a diagram alongside a detailed description, and also now provide formula (8) that quantitatively specifies how the trainable parameters are related to classifier predictions. We would like to work with our reviewers to see what parts of the extra explanations in the appendices can be meaningfully folded back into the main text while not breaking the page count limit.
>
> By the very nature of this work (see 1st response to AnonReviewer3), one would naturally want to study a monochromatic image classification problem, as the experimental physics aspects strongly prefer a monochromatic coherent light source for this sort of task. This suggests using MNIST, but given that there are well-known concerns about the MNIST dataset, we wanted to also explore performance on another dataset that is unrelated but similarly-structured.

---

> ### Author Response · Authors · 2020-11-23
> **Did our explanations and updated article address the reviewer's concerns?**
>
> The ICLR discussion period on this work started 2020-11-10, with publication of the reviewers' assessments. The intent of the discussion period is to facilitate a discussion between authors and reviewers, clarifying questions about a submission. We, the authors, provided a 1st updated version to our paper in a timely fashion, on 2020-11-13, answering the questions of all our reviewers.
>
> We observe that the end of the 2nd stage of the discussion period on 2020-11-24 one day away and the reviewer did not yet provide any feedback on whether the explanations we gave in our response, alongside the expanded explanations we added in appendices to our work, do address their concerns and questions. Tomorrow is the last day for authors to respond to reviewers.
>
> As this may be our last opportunity to address any remaining concerns of the reviewer, we would like to kindly ask the reviewer to please flag up any such concerns with us. Regretfully, remaining time only permits minor changes to the text at this point.

---

### Official Review · AnonReviewer4 · 2020-10-30
**Accessible contribution at intersection of ML and quantum mechanics**

**Rating:** 8
**Confidence:** 3

**Review:**

The paper identifies two atomic problems, respectively in fields of ML (MNIST classification) and quantum mechanics (measuring a single photon), and brings them together in a simplified setup that uses a single photon emitted according to the spatial distribution of images to classify MNIST/Fashion-MNIST. The introduction of quantum mechanics into the problem is through a trainable computational model of a beam splitter/phase shifter mechanism, aka a rotation in a high dimensional complex space, that's allowed to alter the photon's state before hitting the measurement device. The paper shows that using this overly simplified (and claimed to be physically feasible) quantum computer, which acts as the representation learning layer, improves classification accuracy over any other representation learning method that doesn't use quantum computing. The major take-away is an accessible demonstration of how an elementary quantum computer might work for ML, and what may be possible with actual qubits.

Strengths:
* The paper sets out to use two textbook problems in ML and quantum mechanics to introduce a textbook problem at the intersection, and does a fairly good job at analyzing the problem extensively. Given that the overall problem is of broad interest to the representation learning community, the solid execution of the paper is itself a good argument for acceptance.
* Accessible explanation of quantum states, the measurement process, and the building blocks of quantum computing.

Comment:
* The paper analyzes a single problem where no classical representation learning method can improve accuracy due to the fact that there is a single photon. It would be very beneficial to allude to other setups, perhaps in a related work section, to provide broader context and help the reader understand better the significance of the problem.

---

> ### Author Response · Authors · 2020-11-13
> **Added context to updated paper version, as suggested.**
>
> We thank the reviewer for their expert assessment. The updated (likely not yet final) version of our article expands the main text to provide more context about how this work is related to other "analog optics ML" research that focuses on the many-photon case. We strengthened the argument w.r.t. maximal achievable performance by replacing an implicit assumption "for any classifier built without knowledge of the test set" to: "for any classifier, which even might have been built knowing about the structure of the test set". This slightly increased the performance bound on the "classical baseline", but not in a way that would change our argument or conclusions. We also added to the main text formula (8) that concisely summarizes how to obtain a target class probability from an example image.

---

### Decision · Program_Chairs · 2021-01-07
**Final Decision**

**Decision:**

Accept (Poster)

**Comment:**

This is an unusual, but interesting submission. Can we use a simple "quantum computer" (in fact, physical system) to solve classification problems in ML? A single photon passes through the screen. Its state is described by the complex vector. A quantum computer makes a unitary linear transformation on this state in such a way that it maximizes the overlap with a corresponding class. Such a model can be parametrized by conventional means, and trained and later possibly realized by an quantum system

Pros:
1.  The area of QC is very important, and such papers shed a new light on the subject.
2. Inspiration to the ICLR community to work on in this area.
3. Technically correct.


Cons:
1. The accuracies are far from SOTA and use very toy datasets. It is not clear, how to get to the accuracies needed in practice.
2. The actual computational speed of inference is not clear.
3. Discussion of more complicated models and their possibility is necessary.
4. Quite a few misprints are in the text which need to be fixed in the final version.